# Can HER2 1+ Breast Cancer Be Considered as HER2-Low Tumor? A Comparison of Clinicopathological Features, Quantitative HER2 mRNA Levels, and Prognosis among HER2-Negative Breast Cancer

**DOI:** 10.3390/cancers14174250

**Published:** 2022-08-31

**Authors:** Lan Shu, Yiwei Tong, Zhuoxuan Li, Xiaosong Chen, Kunwei Shen

**Affiliations:** Department of General Surgery, Comprehensive Breast Health Center, Ruijin Hospital, School of Medicine, Shanghai Jiaotong University, Shanghai 200025, China

**Keywords:** breast cancer, HER2-low, immunohistochemistry, mRNA level, prognosis

## Abstract

**Simple Summary:**

HER2-low breast cancer is a new entity featuring low HER2 expression (HER2 1+ or HER2 2+/FISH−) and can potentially benefit from novel antibody-drug conjugates. A precise estimation of HER2-low expression is warranted. Here, we found that HER2 1+ and HER2 0 tumors shared similar clinicopathologic features and HER2 mRNA expression, significantly different from HER2 2+/FISH− tumors. A poor concordance rate was found between IHC/FISH and qRT-PCR for the estimation of HER2 expression in HER2-negative tumors. This study suggests that the current definition of HER2-low expression with the lower boundary of HER2 IHC 1+ may be inaccurate. IHC and qRT-PCR were not optimal to quantify HER2-low expression, especially for HER2 1+ patients.

**Abstract:**

Background: Human epidermal growth factor receptor 2 (HER2)-low tumor is a new entity defined as HER2 immunohistochemistry (IHC) 1+ or 2+/fluorescence in situ hybridization (FISH)-negative. We aimed to evaluate whether HER2 mRNA levels tested by quantitative reverse transcriptase-polymerase chain reaction (qRT-PCR) could better define HER2-low tumors. Patients and methods: Consecutive breast cancer patients with hormonal receptor-positive, HER2-negative diseases, and HER2 mRNA results were included. Clinicopathologic features, HER2 mRNA expression level, and prognosis were compared among HER2 0, 1+ and 2+/FISH− groups. Concordance of the HER2 category between qRT-PCR and IHC/FISH was analyzed for each group. Results: 2296 patients were included: 368 (16.0%) HER2 0, 911 (39.7%) 1+, and 1017 (44.3%) 2+/FISH− tumors. HER2 1+ cases shared similarities with HER2 0 tumors in terms of clinicopathologic features (all *p* > 0.05), whereas IHC 2+/FISH− cases were less often non-IDC (*p* = 0.045), node-negative (*p* = 0.044), and Ki-67 < 14% (*p* <0.001). The mRNA expression was similar between HER2 0 and 1+ cases (*p* = 0.063), and both were lower than 2+/FISH− cases (*p* < 0.001). A poor concordance rate was found between IHC/FISH and qRT-PCR for HER2 0 and HER2-low cases (Cohen’s kappa 0.126, *p* < 0.001). No survival difference was observed among these groups, whether stratified by HER2 IHC/FISH status or mRNA level (all *p* > 0.05). Conclusions: HER2 1+ cases had similar clinicopathological features to HER2 0 breast cancers, and both were different from HER2 2+/FISH− cases. HER2 mRNA levels were comparable between HER2 0 and 1+ tumors, and both were significantly lower than IHC 2+/FISH− tumors. Neither IHC nor qRT-PCR may be optimal to quantify HER2-low expression, especially for HER2 1+ patients.

## 1. Introduction

HER2 protein overexpression and/or ERBB2 amplification are present in 15–20% of breast tumors and has been established as a predictive factor of prognosis [1]. With the advent of HER2-targeted therapy, the prognosis of HER2-enriched diseases has been improved in neo-adjuvant, adjuvant, and metastatic settings [2,3]. Thus, accurate determination of HER2 status is critical to identify patients who could benefit from HER2-targeted therapies and to avoid ineffective treatment and unnecessary adverse effects in the management of breast cancer patients [4,5]. 

Currently, immunohistochemistry (IHC) and fluorescence in situ hybridization (FISH) are the standard assays to determine HER2 status. According to the 2018 American Society of Clinical Oncology/College of American Pathologists (ASCO/CAP) guidelines, breast tumors would be considered HER2-positive when IHC 3+ or FISH amplified, whereas HER2-negative is defined as HER2 0, 1+, and 2+ with FISH-negative [6]. However, studies reported significant heterogeneity regarding the HER2 expression within HER2-negative diseases [7,8,9]. Derived from HER2-negative tumors, HER2-low tumors (IHC 1+ or 2+/FISH-negative) characterized by a low HER2 protein level have become a new entity, distinct from HER2 0 tumors in terms of clinicopathologic characteristics, molecular features, and survival [8,10,11]. Meanwhile, recent studies have reported that HER2-low breast cancer might benefit from HER2-targeted antibody-drug conjugates (ADCs) treatment [7,12], such as trastuzumab deruxtecan (T-Dxd) [13] and trastuzumab duocarmazine (SYD985) [14]. Moreover, HER2-low tumors account for approximately 45–55% of breast tumors [7], indicating that a large group of breast cancer patients may benefit from these novel HER2-targeted therapies.

Studies have established interaction between hormonal receptor (HR) status and HER2 expression. Schettini et al. found that, in HER2-low breast tumors, HR-positive cases have higher HER2 mRNA expression compared with HR-negative cases [8]. Clinically, as HER2-low tumors present an inspirable response to novel ADCs, HR-positive breast cancer patients with HER2-low expression may derive benefit from ADCs and may be spared from conventional chemotherapy. Therefore, for the HR-positive HER2-negative population, an accurate detection of HER2 expression through quantitative assays can help select candidates for novel HER2-targeted therapy, changing therapeutic decisions and even prognosis.

As the current standard assays in the determination of HER2-low expression, IHC/FISH have been challenged due to their low accuracy. Lambein et al. reported that 85% of local HER2 IHC 0 cases were scored as 1+ at a central laboratory [15]. Fernandez et al. also reported a low concordance rate of 26% between HER2 0 and 1+ tumors when these cases were evaluated by different pathologists [16]. Thus, IHC may not be the optimal assay to assess HER2-low expression. Meanwhile, many studies reported that quantitative reverse transcriptase-polymerase chain reaction (qRT-PCR) has demonstrated high concordance with IHC/FISH in assessing HER2-positive and -negative status [17,18,19,20], but its relationship in the HER2-low status evaluation was not well-reported.

In this study, we focused on the HER2-negative population and first aimed to compare the differences between HER2 0, HER2 1+, and HER2 2+ FISH-negative tumors in terms of clinicopathologic features, HER2 mRNA expression, and clinical outcomes, so as to demonstrate potential similarity and heterogeneity among these HER2 0 and HER2-low tumors. Moreover, we explored whether testing HER2 mRNA expression by qRT-PCR could be an optimal method for HER2-low disease categorization.

## 2. Materials and Methods

### 2.1. Study Population

We retrospectively included invasive breast cancer patients who received surgery from January 2009 to December 2019 in the Comprehensive Breast Health Center, Ruijin Hospital, Shanghai Jiao Tong University School of Medicine. The eligibility criteria were as follows: (1) HR-positive and HER2-negative breast cancer patients with known 21-gene recurrence score and detailed HER2 mRNA level, or HER2 positive tumors with known HER2 mRNA level, (2) invasive breast cancer, and (3) complete clinicopathological and follow-up data. Patients diagnosed with bilateral breast cancer, de novo stage IV disease, or receiving neoadjuvant treatment were excluded. Those HER2 IHC 2+, without FISH testing, were also excluded from the cohort. Clinicopathological and follow-up details were obtained from the Shanghai Jiao Tong University Breast Cancer Database (SJTU-BCDB). The current study has received approval from the independent Ethical Committees of Ruijin Hospital, Shanghai Jiao Tong University School of Medicine.

### 2.2. Histopathologic Evaluation and FISH-Based Classification

Tumor histopathologic, IHC, and FISH evaluations were conducted by at least two independent, experienced pathologists from the Department of Pathology, Ruijin Hospital, Shanghai Jiao Tong University School of Medicine. The removed tumor specimens were fixed in formalin, embedded in paraffin, performed on 4 μm slides, and stained with primary antibody against HER2 (4B5, Roche) at 42 °C for 16 min in the Ventana BenchMark XT system (Ventana Medical Systems, Tucson, AZ, USA) [21]. HER2 IHC equivocal cases were further assessed using a dual-probe FISH test using PathVysion HER2 DNA kit (Vysis Inc., Downers Grove, IL, USA). Clinical HER2 status was judged according to the 2018 ASCO/CAP guidelines: HER2-negative included HER2 IHC 0, 1+, and 2+ with FISH-negative (average HER2 gene copy number < 4.0 signals/cell regardless of the HER2/CEP17 ratio, or average HER2 gene copy number ≥ 4.0 and <6.0 signals/cell with HER2/CEP17 ratio < 2.0) [6]. The HER2-negative tumors were further categorized into HER2 0, in cases where IHC scored as 0, and HER2-low, defined as IHC scores of 1+ or 2+ with FISH-negative results [7]. HER2-low tumors were classified into HER2 1+ and HER2 2+/FISH subgroups. The methodology and positive criteria for IHC evaluation of ER, PR, and Ki-67 were carried out according to the latest ASCO/CAP guidelines, as presented in our previous work [22,23,24]. HR-positive was defined as ER and/or PR ≥ 1%. According to the 2013 St. Gallen International Expert Consensus, the definition of molecular subtypes was as follows: Luminal A (ER+/HER−, Ki-67 < 14% and PR ≥ 20%)and Luminal B HER2-negative (ER+/HER−, Ki-67 ≥ 14% or ER+/HER−, PR < 20%) [25,26].

In addition, stromal tumor-infiltrating lymphocytes (TILs), stromal CD3, CD4, CD8, and PD-L1 expression were detected in 150 breast tumor samples. Stromal TILs were recorded as the percentage of stromal immune cell infiltration within the tumor boundaries. The counts of CD3+, CD4+, and CD8+ cells in a mm^2^ were recorded for analysis. The median cell counts of CD3+ (3 cell counts/mm^2^), CD4+ (0 cell counts/mm^2^), and CD8+ (12 cell counts/mm^2^) cells were used as cut-off values for analysis. Combined positive score (CPS) was used for the assessment of PD-L1. The definition of CPS was the number of lymphocytes, macrophages, and tumor cells that stained positive for PD-L1 of any intensity relative to total tumor cells. 

### 2.3. HER2 mRNA Expression by qRT-PCR

The expression of HER2 mRNA was measured by qRT-PCR as part of a center-specific 21-gene recurrence (RS) score testing assay using the same formula and genes as OncotypeDX. This RT-PCR-based panel has been validated in large cohorts of Chinese patients and has been routinely applied to guide adjuvant treatment and predict disease outcomes since 2015 [27,28]. In brief, three 10 μm unstained slides of formalin-fixed, paraffin-embedded (FFPE) breast tumor specimens were prepared and reviewed by the pathologists to ensure that invasive tumor components accounted for at least 50% of the section. RNA was extracted using an RNeasy FFPE kit (Qiagen, 73504, Hilden, Germany). Subsequently, reverse transcription was performed using an OmniScript RT kit (Qiagen 205111). PCR was accomplished in the Applied Biosystems 7500 Real-Time PCR System (Foster City, CA, USA) with Premix Ex TaqTM (TaKaRa Bio, RR390A, Shiga, Japan). The expression of HER2 mRNA was measured by cycle threshold (CT). The CT valve was the number of amplification cycles during the qRT-PCR amplification process, when the fluorescence signal of the amplification product reached the set fluorescence threshold. Thus, higher CT values mean lower initial RNA levels and lower CT values mean higher initial RNA levels. The HER2 CT value was adjusted by five endogenous reference genes—β-actin, GAPDH, GUS, RPLPO, and TFRC—and was recorded as ΔCT = CT _HER2_ − CT _Reference_. Forty-eight patients with heterogeneous HER2-positive disease or HER2 IHC 3+ with FISH-negative were considered HER2-positive and included as positive control. In these circumstances, 21-gene RS testing would be recommended to test the HER2 mRNA level for reference and to provide genetic data to guide adjuvant treatment decision.

### 2.4. Follow-Up

Patient follow-up information was recorded annually by specialized breast cancer nurses at our center. The definition of endpoint was referred to the latest standardized definitions for efficacy endpoints (STEEP) criteria [29]. Disease-free survival (DFS) was defined as the period from the date of surgery to the first proven event, including local-regional recurrence, distant metastasis, contralateral breast cancer, second non-breast primary cancer, and death from any cause. Overall survival (OS) was defined as the period from the date of surgery to death from any cause. The last follow-up was finished in January 2022.

### 2.5. Statistical Analysis

Categorical data were analyzed using a chi-squared test (Fisher’s exact test, when necessary) and multinomial logistic regression analysis. The Kruskal–Wallis test was applied to compare HER2 mRNA expression levels by different HER2 IHC/FISH statuses. Receiver operator characteristic (ROC) curves were used for obtaining the optimal ΔCT cut-off value at the most extensive Youden index. Cohen’s kappa statistics were used for evaluating concordance between IHC/FISH and mRNA level by qRT-PCR in determining HER2 status. The comparison of continuous HER2 mRNA level among different HER2 IHC/FISH statuses was shown in the smoothed density plot. Survival curves were performed by Kaplan–Meier analysis and log-rank test. All statistical analyses were carried out by SPSS version 24.0 (SPSS, Inc., Chicago, IL, USA), GraphPad Prism9.0 (GraphPad Software, San Diego, CA, USA), and R software (version 4.0.5, R Foundation for Statistical Computing, Vienna, Austria). All statistical tests were two-tailed and were considered statistically significant at *p*-values < 0.05.

## 3. Results

### 3.1. Baseline Characteristics

A total of 2296 patients with HR-positive HER2-negative breast cancer were ultimately included in the study (Figure 1). In addition, 48 HER2-positive breast cancer patients with known HER2 mRNA levels were included. Among these HER2-negative patients, 368 (16.0%) were HER2 0, 911 (39.7%) were HER2 1+, and 1017 (44.3%) were HER2 2+/FISH−. The median age was 60 years old (range, 24–93).

Histology, TNM stage, tumor size, node status, PR, Ki-67, and molecular subtype were differently distributed among HER2 0, HER2 1+, and HER2 2+/FISH− in the univariate model (all *p* < 0.05; Table 1). Multinomial logistic regression showed a significantly distinguishable distribution of histology (*p* = 0.015), node status (*p* = 0.016), and Ki-67 (*p* = 0.025) among HER2 0, HER2 1+, and HER2 2+/FISH− cases (Table 2). Compared with patients in the HER2 2+/FISH− group, HER2 0 patients were more often non-IDC (odds ratio [OR] 1.57, 95% confidence interval [CI] 1.14–2.15, *p* = 0.006), node-negative (OR 1.58, 95% CI 1.11–2.25, *p* = 0.012), PR-negative (OR 1.54, 95% CI 1.08–2.20, *p* = 0.017); HER2 1+ patients were more often non-IDC (OR 1.30, 95% CI 1.01–1.67, *p* = 0.045), node-negative (OR 1.29, 95% CI 1.01–1.65, *p* = 0.044), and more likely to have Ki-67 < 14% (OR 1.42, 95% CI 1.18–1.70, *p* < 0.001). However, there was no statistically significant difference between HER2 0 and HER2 1+ groups in terms of the above clinicopathologic features (all *p* > 0.05).

Additionally, no significant difference was observed in the distribution of stromal TILs, CD3, CD4, CD8, and PD-L1 among different HER2 IHC/FISH groups (all *p* > 0.05; Appendix A). 

As to the adjuvant systemic treatment, 46.5% of HER2 0, 44.8% of HER2 1+, and 55.3% of HER2 2+/FISH− patients received chemotherapy (*p* < 0.001). No statistically significant differences among these groups in terms of radiation therapy and endocrine therapy were observed (all *p* > 0.05; Appendix A).

### 3.2. HER2 mRNA Expression Comparison by HER2 IHC/ FISH Status

HER2 mRNA expression was determined by qRT-PCR and recorded as ΔCT (CT_HER2_− CT _reference_). The medians of ΔCT were 2.94 (*n* = 368; Q1–Q3: 2.25–3.57), 2.61 (*n* = 1928; Q1–Q3: 1.90–3.23), and 1.18 (*n* = 48; Q1–Q3: 0.24–1.90) for HER2 0, HER2-low, and HER2-positive cases, respectively (Figure 2A). HER2-positive cases showed the highest mRNA expression level while HER2 0 showed the lowest mRNA expression level, and all pairwise comparisons were significantly different (*p* < 0.001; Figure 2A). When further stratified by IHC score in HER2-low tumors, the medians of ΔCT were 2.80 (*n* = 911; Q1–Q3: 2.16–3.46) and 2.47 (*n* = 1017; Q1–Q3: 1.74–3.03) for HER2 1+ and HER2 2+/FISH− cases, respectively (Figure 2B). We found that HER2 mRNA expression of HER2 1+ cases was similar to that of HER2 0 cases (*p* = 0.063), and both were much lower than HER2 2+/FISH− cases (*p* < 0.001; Figure 2B). Significant overlaps were shown between HER2 0 and HER2 1+ groups, relatively distinct from HER2 2+/FISH− group (Appendix A). 

### 3.3. Concordance between IHC/FISH and qRT-PCR for HER2 Assessment

We next examined whether HER2 mRNA better categorized patients compared with the conventional IHC/ FISH. ROC curves were applied to determine the optimal ΔCT cut-off for mRNA, which was set to 3.10 in HER2 0 and low cases (area under the curve [AUC] 0.603, *p* < 0.001). The HER2 mRNA low level was defined as ΔCT ≥ 3.10 and the high level as ΔCT < 3.10 (Appendix A). Of the 1928 HER2-low patients, 1445 patients were classified as mRNA high level and 483 patients as mRNA low level. Among 368 HER2 0 patients, 216 and 152 patients were classified as mRNA high and low levels, respectively. HER2 mRNA expression level showed significant discordance compared with the HER2 IHC/FISH status in HER2-low and HER2 0 cases (Cohen’s kappa 0.126, *p* < 0.001; Figure 3A). The concordance rates of HER2-low and HER2 0 cases were 74.9% (1445/1928) and 41.3% (152/368), respectively. 

As HER2 1+ cases were similar to HER2 0 cases in terms of clinicopathological features and HER2 mRNA level, we further combined HER2 0 and HER2 1+ cases as the HER2 0/1+ group to compare with HER2 2+/FISH− cases. The ΔCT cut-off was set to 2.98 in HER2 0/1+ and HER2 2+/FISH− cases (AUC 0.625, *p* < 0.001). The HER2 mRNA expression low level was defined as ΔCT ≥ 2.98 and the high level as ΔCT < 2.98 (Appendix A). Of the 1017 HER2 2+/FISH− patients, 789 and 228 patients were categorized as mRNA high and low levels, respectively. Among the 1279 HER2 0/1+ patients, 764 and 515 patients were categorized as mRNA high and low levels, respectively. Significant HER2 mRNA expression level discordance was observed between HER2 2+/FISH− and HER2 0/1+ cases (Cohen’s kappa 0.169, *p* < 0.001; Figure 3B). The concordance rates of HER2 2+/FISH− and HER2 0/1+ were 77.6% (789/1017) and 40.3% (515/1279), respectively.

### 3.4. Clinical Outcomes

The median follow-up time for the overall population was 55.8 (range: 12.3–151.8) months. No statistically significant differences in terms of DFS and OS were observed between HER2 0 and HER2-low groups, nor among HER2 0, HER2 1+, and HER2 2+/FISH− groups (all *p* > 0.05; Figure 4). 

We further explored whether HER2 mRNA expression levels could better predict prognosis. The HER2 mRNA expression cut-off (ΔCT 3.10) based on HER2 0 and HER2-low cases was selected to divide these cases into mRNA low and high levels. No statistically significant differences were observed in terms of DFS and OS between the two groups (all *p* > 0.05; Appendix A). Similar results were also obtained when the whole population was categorized by both IHC status and mRNA expression level as four groups (all *p* > 0.05; Appendix A). We further classified the whole population into chemotherapy-treated and chemotherapy-free cohorts. In chemotherapy-free cohorts, DFS showed no significant difference among HER2 0, 1+ and 2+/FISH− groups (*p* > 0.05; Appendix A), while OS in HER2 2+/FISH− cases was significantly better than HER2 0 and 1+ cases (*p* = 0.014; Appendix A). In chemotherapy-treated cohorts, DFS and OS were similar among the three groups (all *p* > 0.05; Appendix A).

Univariate analysis identified that TNM stage, tumor size, histological grade, PR, Ki-67, and chemotherapy were associated with DFS (all *p* < 0.05; Appendix A). TNM stage, tumor size, breast surgery, and histological grade were associated with OS (all *p* < 0.05; Appendix A). Further multivariate Cox regression analysis identified tumor size and histological grade as independent prognostic factors: Tumor size < 2 cm was associated with better DFS and OS, with hazard ratio (HR) 0.55 (95% CI 0.40–0.76, *p* < 0.001), and HR 0.48 (95% CI 0.25–0.91, *p* = 0.024), respectively; tumor grade I-II was associated with better OS, with HR 0.49 (95% CI 0.24–0.97, *p* = 0.041; Appendix A).

## 4. Discussion

HER2-negative tumors are a group of heterogeneous diseases. The precise classification of HER2 status is of great importance in the management of breast cancer patients. With the development of HER2-targeted ADCs, they have shown promising antitumor efficacy in HER2-low expression tumors. However, the optimal assays for HER2-low tumors need to be further studied due to their low concordance rate between HER2 0 and HER2 1+ status evaluation [16]. In this study, a large number of HR-positive HER2-negative patients were included with known HER2 mRNA levels. We found that HER2 0 and HER2 1+ cases shared great similarities in terms of clinicopathological features which were differently distributed compared with HER2 2+/FISH− cases. More importantly, our study demonstrated that the HER2 mRNA expression levels were similar between HER2 0 and HER2 1+ cases, both lower than HER2 2+/FISH− cases, which may explain the low concordance rate between HER2 0 and HER2 1+ tumors. Another important result was that the discordance rate was high between IHC/FISH and qRT-PCR in determining HER2 status in HR-positive and HER2-negative tumors. To our knowledge, this is the first large consecutive cohort focused on HER2-negative breast cancer population to systematically explore the accuracy of IHC/FISH or qRT-PCR methods to detect the HER2-low expression and define the optimal HER2-low breast tumors.

HER2-low tumors account for approximately 45–55% of breast tumors [7], while 64.5% of HER2-low breast tumors were HR-positive [8]. Numerous studies have shown that novel ADCs have hopeful antitumor activity in HR-positive and HR-negative breast cancer with HER2-low expression [12,30]. Despite endocrine therapy as the optimal management, a proportion of HR-positive breast cancer patients will still relapse, and novel ADCs may offer a potential effective treatment in these HR+/HER2-low tumors to improve survival. The DESTINY-Breast04 study (NCT03734029) showed a superior PFS with T-Dxd to the investigator’s choice chemotherapy in HR+/HER2-low breast cancer with at least two lines of endocrine therapy in a metastatic setting [31,32]. A phase III clinical trial (NCT04494425/DESTINY-Breast06), exclusively enrolling patients with HR-positive/HER2-low metastatic breast cancer, is ongoing to evaluate the efficacy of T-Dxd compared with chemotherapy [33]. Therefore, it is of important clinical significance to accurately define HER2-low expression in HR-positive HER2-negative population.

DAISY (NCT04132960), a phase II study enrolling only HER2 status breast cancer patients, reported that the median progression-free survival (mPFS) for HER2-positive cohort, HER2-low cohort, and HER2 0 cohort was 11.1 months, 6.7 months, and 4.2 months, respectively; this showed that the antitumor activity of T-Dxd correlated with HER2 expression level exhibiting a better response rate in higher HER2 expression tumors, indicating that HER2 expression level can be recognized as a possible marker of efficacy to T-Dxd [34]. However, the mPFS of HER2-low cohort (*n* = 72) in the DAISY study was lower than in a phase Ib study (*n* = 54) [13,35], which may relate to the different sample sizes or detection of HER2 status. In addition to T-Dxd, other novel ADCs, such as RC48 and SYD985, also have shown promising antitumor activity in HER2-low breast tumors [14,36]. With the development of these novel HER2-targeted ADCs, there is an urgent need to precisely define HER2 expression level. However, as the HER2 testing assays were initially developed for their accurate positivity or negativity definition [16,37,38], there is no certain definition for HER2-low tumors. Although the upper boundary of HER2-low as HER2 2+ with FISH-negative is clear, whether HER2 1+ can be the lower boundary lacks a gold standard [39], as high heterogeneity of HER2 IHC staining exists in HER2 1+ cases [40]. Thus, the current unmet medical need is to define the optimal cut-off value for HER2 0 and HER2-low tumors with sensitive assays, which will provide more opportunities for these HER2-low patients to receive novel HER2-targeted therapy.

IHC is a semiquantitative assay and may be influenced by observer variability and other factors [7,20]. Meanwhile, low concordance rates were found in the evaluation of HER2 IHC status in these patients with HER2 0 or HER2 1+ tumors [15,16]. Thus, when HER2 0 and 1+ cases are not easily distinguished by IHC, the question is whether HER2-low tumors can be managed as a biologically different entity. In our study, HER2 0 and HER2 1+ cases shared great similarities in terms of clinicopathologic features as well as HER2 mRNA expression, yet both were distinct from HER2 2+/FISH− cases, which may further indicate difficulty in accurate evaluation of HER2 0 and 1+ breast tumors. Overall, it may be inappropriate to classify HER2 IHC 1+ as HER2-low expression, and more sensitive assays are necessary for further accurately assessing HER2-low status, especially for HER2 1+ patients. 

Noske et al. reported that standardized quantitative methods such as qRT-PCR are preferable for the evaluation of HER2 status [20]. qRT-PCR could present the true continuum of HER2 mRNA expression levels and has been assessed as a potential alternative to IHC/FISH, as qRT-PCR enables reproducible quantification of HER2 mRNA expression and reduces observer variability [7,18]. In the current study, we found a poor concordance rate between IHC/FISH and qRT-PCR for HER2 status evaluation in HR-positive HER2-negative cases. This is consistent with the results reported by Xu et al.: discordance exists between IHC and mRNA levels in determining HER2-low status in the whole HER2-negative population [41]. Upon further analysis, we discovered that, compared with HER2 2+/FISH− cases, lower concordance rates were found between IHC score and mRNA level in HER2 0 or HER2 0/1+ cases, indicating that the HER2 mRNA distribution in HER2 0 and HER2 1+ cases was chaotic. Thus, HER2 IHC 1+ may not be adequate for HER2-low tumor definition. Further biomarkers to better define HER2-low tumors are necessary and novel anti-HER2 ADCs treatment in these patients should be explored with further biomarker analysis. 

Several limitations of our study require attention. Firstly, the study was retrospectively designed and carried out in a single center, which could cause certain selection bias. Additionally, forty-eight patients with heterogeneous HER2-positive disease or HER2 IHC 3+ with FISH−negative were considered HER2-positive and included in our study as positive control. These HER2-positive patients were not randomly selected which might cause selection bias. Additionally, only HR-positive HER2-negative breast cancer patients were included for HER2-low population analysis in this cohort. However, in the HER2-low breast tumors, it has been reported that HR-negative cases have lower HER2 mRNA levels than HR-positive cases [8]. Moreover, a phase Ib clinical trial of T-Dxd reported that the ORR seemed to be different according to HR status (HR-positive 40.4% vs. HR-negative 14.3%) [13]. Therefore, in HR-negative cases, further research is merited to explore the characteristics of HER2-low tumors and to find out whether HER2 1+ and HER2 2+/FISH− tumors can be considered an entity.

## 5. Conclusions

Our study found that HER2-low patients with HER2 1+ tumors were more similar to HER2 0 tumors, which have diverse clinicopathologic factors and HER2 mRNA levels compared with HER2 2+/FISH− cases, indicating the current definition of HER2-low expression with the lower boundary of HER2 IHC 1+ may be inaccurate. Neither IHC nor qRT-PCR may be the optimal assay to quantify HER2-low population. Trials integrating quantitative qRT-PCR testing of HER2 mRNA levels and response to anti-HER2 ADCs treatment warrant further evaluation.

## Figures and Tables

**Figure 1 cancers-14-04250-f001:**
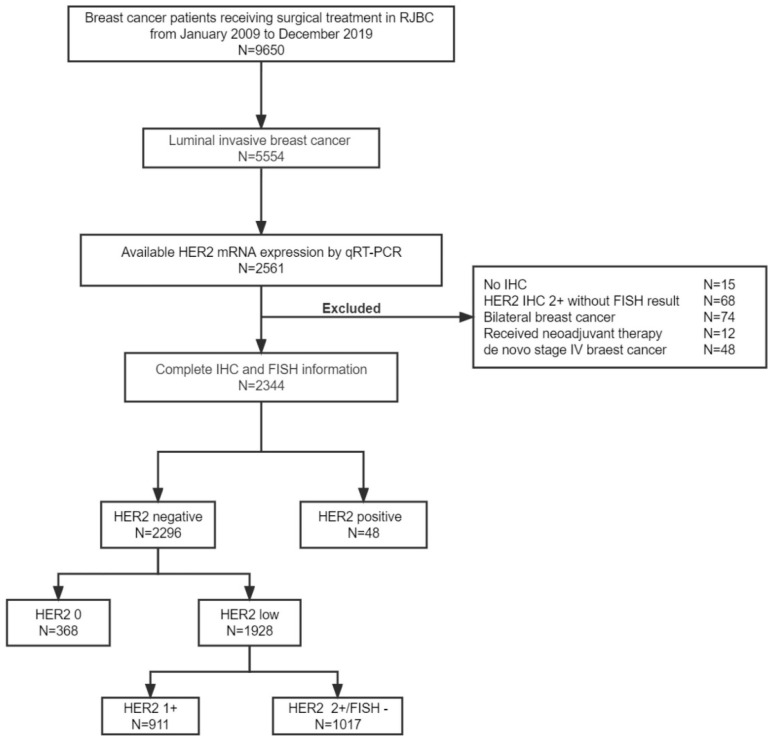
Study flowchart. Abbreviations: HER2, human epidermal growth factor receptor 2; mRNA, message ribonucleic acid; qRT-PCR, quantitative reverse transcriptase-polymerase chain reaction; IHC, immunohistochemistry; FISH, fluorescence in situ hybridization.

**Figure 2 cancers-14-04250-f002:**
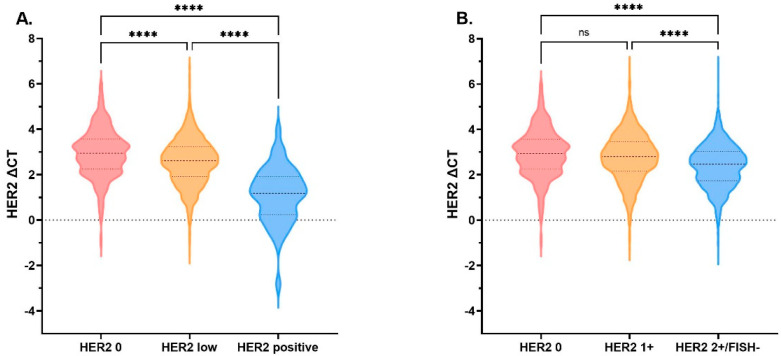
HER2 mRNA expression by HER2 IHC/FISH status. HER2 mRNA expression tested by qRT-PCR assay, adjusted by endogenous genes β-actin, GAPDH, GUS, RPLPO, and TFRC. (**A**) The medians of ΔCT were 2.94 (*n* = 368; Q1–Q3: 2.25–3.57), 2.61 (*n* = 1928; Q1–Q3: 1.90–3.23), and 1.18 (*n* = 48; Q1–Q3: 0.24–1.90) for HER2 0, HER2-low, and HER2-positive cases, respectively. (**B**) The medians of ΔCT were 2.94 (*n* = 368; Q1–Q3: 2.25–3.57), 2.80 (*n* = 911; Q1–Q3: 2.16–3.46), and 2.47 (*n* = 1017; Q1–Q3: 1.74–3.03) for HER2 0, HER2 1+, and HER2 2+/FISH− cases, respectively. No statistically significant difference in HER2 mRNA expression was detected between HER2 0 and HER2 1+ cases (*p*= 0.063); however, HER2 expression of HER2 2+/FISH− was significantly higher than HER2 0 (*p* < 0.0001) and HER2 1+ (*p* < 0.0001). **** *p* < 0.0001. The thick dotted line denotes the median, the thin dotted line denotes inter-quartile range. Abbreviations: HER2, human epidermal growth factor receptor 2; mRNA, message ribonucleic acid; IHC, immunohistochemistry; FISH, fluorescence in situ hybridization; qRT-PCR, quantitative reverse transcriptase-polymerase chain reaction; CT, cycle threshold; ns, not significant.

**Figure 3 cancers-14-04250-f003:**
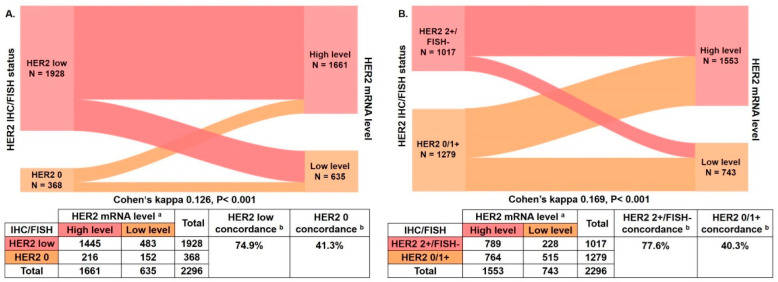
Concordance between HER2 IHC/FISH status and HER2 mRNA levels. (**A**) The percent concordances of HER2-low and HER2 0 were 74.9% and 41.3%, respectively, and Cohen’s kappa was 0.126, *p* < 0.001; (**B**) The percent concordances of HER2 2+/FISH− and HER2 0/1+ were 77.6% and 40.3%, respectively, and Cohen’s kappa was 0.169, *p* < 0.001. ^a^ HER2 mRNA expression level was classified as low level or high level by the ΔCT cut-off value from the ROC curve; the ΔCT cut-off between HER2 0 and HER2-low was 3.10 (low level, ΔCT ≥ 3.10; high level, ΔCT < 3.10); the ΔCT cut-off between HER2 0/1+ and HER2 2+/FISH− was 2.98 (low level, ΔCT ≥ 2.98; high level, ΔCT < 2.98). ^b^ Concordance was calculated using IHC/FISH as standard test. Abbreviations: HER2, human epidermal growth factor receptor 2; IHC, immunohistochemistry; FISH, fluorescence in situ hybridization; mRNA, message ribonucleic acid; CT, cycle threshold.

**Figure 4 cancers-14-04250-f004:**
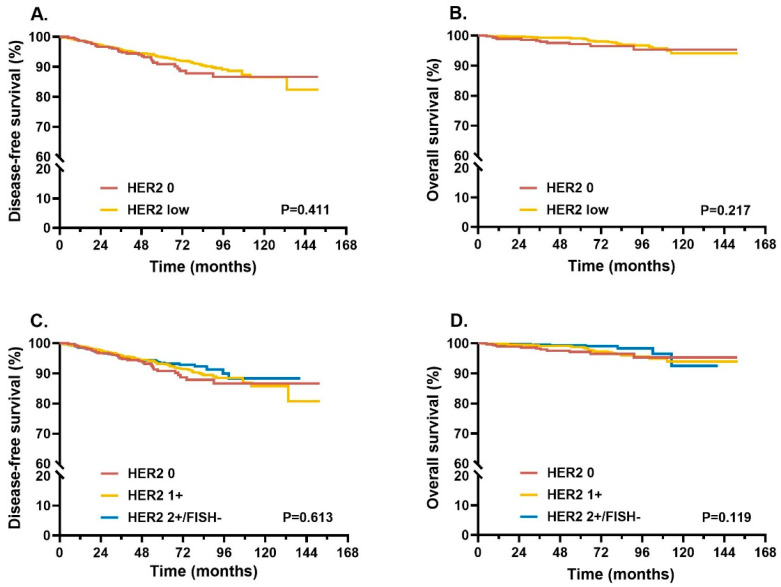
Survival in different HER2 IHC/FISH status patients. Kaplan–Meier curves of DFS (**A**) and OS (**B**) were compared between HER2 0 and HER2-low patients. Further comparisons of DFS (**C**) and OS (**D**) were performed in HER2 0, HER2 1+, and HER2 2+/FISH− patients. *p*-values for log-rank tests were reported at the bottom of each box. Abbreviations: HER2, human epidermal growth factor receptor 2; IHC, immunohistochemistry; FISH, fluorescence in situ hybridization; DFS, disease-free survival; OS, overall survival.

**Table 1 cancers-14-04250-t001:** Baseline patient characteristics by HER2 IHC/FISH status.

Characteristics	Total *n* = 2296 (%)	HER2 0 *n* = 368 (%)	HER2 1+ *n* = 911 (%)	HER2 2+/FISH− *n* = 1017 (%)	*p*-Value
Age (y/o)					0.798
<60	1270 (55.3)	204 (55.4)	511 (56.1)	555 (54.6)	
≥60	1026 (44.7)	164 (44.6)	400 (43.9)	462 (45.4)	
Gender					0.806
Female	2277 (99.2)	366 (99.5)	903 (99.1)	1008 (99.1)	
Male	19 (0.8)	2 (0.5)	8 (0.9)	9 (0.9)	
Histology					0.004
IDC	1935 (84.3)	295 (80.2)	756 (83.0)	884 (86.9)	
Non-IDC	361 (15.7)	73 (19.8)	155 (17.0)	133 (13.1)	
TNM stage					0.020
I	1423 (62.0)	242 (65.8)	588 (64.5)	593 (58.3)	
II	859 (37.4)	123 (33.4)	317 (34.8)	419 (41.2)	
III	14 (9.6)	3 (0.8)	6 (0.7)	5 (0.5)	
Tumor Size					0.050
<2 cm	1227 (53.4)	201 (54.6)	511 (56.1)	515 (50.6)	
≥2 cm	1069 (46.6)	167 (45.4)	400 (43.9)	502 (49.4)	
Breast surgery					0.372
BCS	1006 (43.8)	166 (45.1)	411 (45.1)	429 (42.2)	
Mastectomy	1290 (56.2)	202 (54.9)	500 (54.9)	588 (57.8)	
ALN status					0.001
Negative	1919 (83.6)	323 (87.8)	777 (85.3)	819 (80.5)	
Positive	377 (16.4)	45 (12.2)	134 (14.7)	198 (19.5)	
Histological grade					0.334
I-II	1879 (82.8)	292 (79.3)	755 (82.9)	832 (81.8)	
III	417 (18.2)	76 (20.7)	156 (17.1)	185 (18.2)	
ER					0.115
Negative	5 (0.2)	1 (0.3)	4 (0.4)	0 (0.0)	
Positive	2291 (99.8)	367 (99.7)	907 (99.6)	1017 (100.0)	
PR					0.037
Negative	250 (10.9)	54 (14.7)	95 (10.4)	101 (9.9)	
Positive	2046 (89.1)	314 (85.3)	816 (89.6)	916 (90.1)	
Ki-67					<0.001
<14%	1065 (46.4)	178 (48.4)	467 (51.3)	420 (41.3)	
≥14%	1231 (53.6)	190 (51.6)	444 (48.7)	597 (58.7)	
Molecular subtype					0.010
Luminal A	733 (31.9)	119 (32.3)	321 (35.2)	293 (28.8)	
Luminal B (HER2-)	1563 (68.1)	249 (67.7)	590 (64.8)	724 (71.2)	

Abbreviations: HER2, human epidermal growth factor receptor 2; IHC, immunohistochemistry; FISH, fluorescence in situ hybridization; y/o, years old; IDC, invasive ductal carcinoma; BCS, breast-conserving surgery; ALN, axillary lymph node; ER, estrogen receptor; PR, progesterone receptor.

**Table 2 cancers-14-04250-t002:** Multivariate analysis of patient characteristics by HER2 IHC/FISH status ^a^.

Characteristics	HER2 0 *n* = 368			HER2 1+ *n* = 911				*p*
	OR	95% CI	*p*	OR	95% CI	*p*	*p* ^b^	
Histology			0.006			0.045	0.233	0.015
Non-IDC	1.57	1.14–2.15		1.30	1.01–1.67			
IDC	1.00			1.00				
Tumor Size			0.395			0.110	0.730	0.265
<2 cm	1.11	0.87–1.42		1.16	0.96–1.40			
≥2 cm	1.00			1.00				
ALN status			0.012			0.044	0.264	0.016
Negative	1.58	1.11–2.25		1.29	1.01–1.65			
Positive	1.00			1.00				
PR			0.017			0.668	0.083	0.073
Negative	1.54	1.08–2.20		1.07	0.79–1.44			
Positive	1.00			1.00				
Ki-67			0.070			<0.001	0.319	0.025
<14%	1.25	0.98–1.60		1.42	1.18–1.70			
≥14%	1.00			1.00				
Molecular subtype			0.452			0.691	0.686	0.948
Luminal A	0.87	0.59–1.26		0.94	0.71–1.26			
Luminal B (HER2-)	1.00			1.00				

^a^ The reference category for subtype characteristics is HER2 2+/FISH−; ^b^ Refers to HER2 0 vs. HER2 1+; Abbreviations: HER2, human epidermal growth factor receptor 2; IHC, immunohistochemistry; FISH, fluorescence in situ hybridization; OR, odds ratio; CI, confidence interval; IDC, invasive ductal carcinoma; ALN, axillary lymph node; PR, progesterone receptor.

## Data Availability

The data presented in this study are available in this article (and Appendix A).

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
