# Peer review of "Can HER2 1+ Breast Cancer Be Considered as HER2-Low Tumor? A Comparison of Clinicopathological Features, Quantitative HER2 mRNA Levels, and Prognosis among HER2-Negative Breast Cancer"

_cancers, 2022, doi:10.3390/cancers14174250_

Round 1

Reviewer 1 Report (Previous Reviewer 3)

-

Reviewer 2 Report (Previous Reviewer 1)

The authors showed that HER2 1+ breast cancer cases had similar clinical and pathological features with HER2 0 breast cancer, both differing from HER2 2+/FISH- cases. HER2 mRNA levels were comparable between HER2 0 and 1+ tumors, both significantly lower than in IHC 2+/FISH- tumors. The authors suggested that either IHC or qRT-PCR may not be optimal for quantifying low HER2 expression, especially in patients with HER2 1+. The authors submitted the article for consideration again, however, they took into account and reasonably commented on most of the comments of the reviewers. I think the article is interesting and able to attract the attention of readers. I recommend accepting the article in its present form.

This manuscript is a resubmission of an earlier submission. The following is a list of the peer review reports and author responses from that submission.

Round 1

Reviewer 1 Report

The authors of the article sought to evaluate whether quantitative HER2 mRNA levels tested by reverse transcriptase polymerase chain reaction (qRT-PCR) could better identify low HER2 tumors. It was shown that HER2 1+ cases were similar to HER2 0 tumors in terms of clinical and pathological features (all P>0.05), while IHC 2+/FISH- cases differed. The level of mRNA expression was similar in the HER2 0 and 1+ cases (P = 0.063), both lower than in the 2+/FISH- cases. This is an interesting work that will undoubtedly be useful for the selection and planning of treatment tactics for patients with breast cancer.

There are several questions for the authors:

1. It is not indicated what kind of treatment besides surgery the patients received. This is especially important when evaluating survival. Thus, patients with the same HER2 status but different treatments (as well as different clinicopathological characteristics) may have a different prognosis. It is correct to compare only patients with the same treatment regimens.

2. There is no information about the stage of breast cancer according to TNM, it is only indicated that stage 4 was excluded. But in table 1, these data must be added.

3. Why did the authors not consider the HER2 3+ group for comparison? I think it would be good for clarity.

Author Response

Point 1: It is not indicated what kind of treatment besides surgery the patients received. This is especially important when evaluating survival. Thus, patients with the same HER2 status but different treatments (as well as different clinicopathological characteristics) may have a different prognosis. It is correct to compare only patients with the same treatment regimens.

Response 1:

Thank you for your suggestion. The adjuvant treatment information was listed in Supplement Table S1, and also stated in the last paragraph of “3.1 Baseline characteristics” as follows: As to the adjuvant systemic treatment, 46.5% HER2 0, 44.8% HER2 1+, and 55.3% HER2 2+/FISH- patients received chemotherapy (P<0.001). No statistically significant differences among these groups in terms of radiation therapy and endocrine therapy were observed (All P>0.05, Supplement Table S1).

We fully agree with your opinion that clinicopathological and treatment bias should be adjusted in survival analysis. Thus, we have taken your advice and add the prognosis comparison between HER2 subgroups in chemotherapy-treated or chemotherapy-free cohorts as Figure S4. We have also provided univariate analysis of prognostic factors as Supplement Table S2 and Cox regression analysis of prognostic factors as Supplement Table S3. (as shown in the attached Word)

Point 2: There is no information about the stage of breast cancer according to TNM, it is only indicated that stage 4 was excluded. But in table 1, these data must be added.

Response 2: Thank you for your advice. We have taken your advice and added the TNM stage in the Table 1. As tumor size and axillary lymph node status showed strong collinearity with TNM stage, we did not include the TNM stage in the multinomial logistic regression analysis.(as shown in the attached Word)

Point 3: Why did the authors not consider the HER2 3+ group for comparison? I think it would be good for clarity.

Response 3: Thank you for your comment. Aleix Prat et al. have reported that ERBB2 mRNA level in HER2 3+ tumors was significantly higher than HER2 2+ tumors (P<0.001), meanwhile, the ERBB2 mRNA level in HER2 2+/FISH+ tumors was significantly higher than HER2 2+/FISH- tumors(P<0.001)[2,3]. Here, since our study was focused exclusively on HER2-low tumors, most of the included cases were HER2-low or HER2 0 tumors. HER2-positive tumors were only a positive control. We fully agree with you that HER2 3+ cases might be a better positive control. However, on the one hand mRNA data were limited for HER2 3+ cases (n=48) in our database, and on the other hand the mRNA level were significantly higher than tumors in other HER2 IHC status. Therefore, the inclusion of HER2 3+ cases might bring statistical bias. In future studies, we will try to supplement HER2 3+ mRNA data for further analysis.

Reviewer 2 Report

  The current manuscript described a study aiming to define the clinical relevance of varying expression of Her2 oncogene in the Her2- breast cancer patient population.  At a first glance, the study aim appears quite odd, as it appeared to explore the clinical importance of Her2 expression in the Her2-negaive patient population. It is kind of reminiscent of planning a journey trying to catch fish in a pond with nearly depleted fish resources. As a result, there is of little surprise that the output from the study was highlighted by lack of correlation between Her2 expression and patient survival. While the reviewer admires the hard work and careful efforts of the authors, the negative result may have limited appeal to researchers and clinicians in the breast cancer research field.

Author Response

Point 1: The current manuscript described a study aiming to define the clinical relevance of varying expression of Her2 oncogene in the Her2- breast cancer patient population.  At a first glance, the study aim appears quite odd, as it appeared to explore the clinical importance of Her2 expression in the Her2-negaive patient population. It is kind of reminiscent of planning a journey trying to catch fish in a pond with nearly depleted fish resources. As a result, there is of little surprise that the output from the study was highlighted by lack of correlation between Her2 expression and patient survival. While the reviewer admires the hard work and careful efforts of the authors, the negative result may have limited appeal to researchers and clinicians in the breast cancer research field.

Response 1:

Thank you for your comment.

HER2-low tumor is a new entity defined as HER2 1+ or 2+ with FISH negative. Currently, lots of studies have reported that novel antibody-drug conjugates (ADC) showed hopeful anti-tumor activity in HER2-low breast cancer population, which provided a promising treatment choice to improve survival in HER2-low patients. However, the mechanism of action of ADC requires certain expression of HER2 protein in tumor cells. The current HER2 testing algorithm was initially designed to distinguish “HER2 positive” from “HER2 negative” [1-3], but was not capable of quantifying the lower boundary of HER2-low expression. As a result, it is quite crucial for us to define the optimal HER2 expression level (zero vs. low) in HER2-negative tumors to select candidates for ADC treatment. This is the main aim of our study.

We found that HER2 0 and HER2 1+ cases shared great similarities in terms of clinicopathological features, which were differently distributed compared with HER2 2+/FISH- cases. More importantly, our study demonstrated that the HER2 mRNA expression levels were similar between HER2 0 and HER2 1+ cases, both lower than HER2 2+/FISH- cases, indicating current lower boundary of HER2-low expression evaluated by IHC/FISH may be inappropriate. Meanwhile, several studies are undergoing to explore how to accurately define HER2-low expression. Myrto Moutafi et al.[4] have redesigned an assay as the AQUA™ method of quantitative immunofluorescence to test a range of antibody concentrations within the lower range of HER2 expression, which may help to  select candidates for T-DXd or similar anti-HER2 conjugated ADCs. In the future, more precise assays to detect HER2-low expression were essential.

The other aim of our study was to look for a better technique to examine HER2 level in this group of patients with HER2-negative tumors. We found that the discordance rates were high between IHC/FISH and qRT-PCR in determining HER2 low status. Thus, our study is of clinical implications as we demonstrate that either IHC or qRT-PCR might not be optimal to quantify HER2-low expression, especially for HER2 1+ patients. More researches are necessary for precise definition of HER2-low expression.

In addition, we have checked and corrected the language mistakes for the revised manuscript. Further English modification will also be provided by editorial office.

Reference

  1. Fernandez, A.I.; Liu, M.; Bellizzi, A.; Brock, J.; Fadare, O.; Hanley, K.; Harigopal, M.; Jorns, J.M.; Kuba, M.G.; Ly, A.; et al. Examination of Low ERBB2 Protein Expression in Breast Cancer Tissue. JAMA Oncol 2022, 8, 1-4, doi:10.1001/jamaoncol.2021.7239.
  2. Ruschoff, J.; Lebeau, A.; Kreipe, H.; Sinn, P.; Gerharz, C.D.; Koch, W.; Morris, S.; Ammann, J.; Untch, M.; Nicht-interventionelle Untersuchung, H.E.R.S.G. Assessing HER2 testing quality in breast cancer: variables that influence HER2 positivity rate from a large, multicenter, observational study in Germany. Mod Pathol 2017, 30, 217-226, doi:10.1038/modpathol.2016.164.
  3. Pfitzner, B.M.; Lederer, B.; Lindner, J.; Solbach, C.; Engels, K.; Rezai, M.; Dohnal, K.; Tesch, H.; Hansmann, M.L.; Salat, C.; et al. Clinical relevance and concordance of HER2 status in local and central testing-an analysis of 1581 HER2-positive breast carcinomas over 12 years. Mod Pathol 2018, 31, 607-615, doi:10.1038/modpathol.2017.171.
  4. Moutafi, M.; Robbins, C.J.; Yaghoobi, V.; Fernandez, A.I.; Martinez-Morilla, S.; Xirou, V.; Bai, Y.; Song, Y.; Gaule, P.; Krueger, J.; et al. Quantitative measurement of HER2 expression to subclassify ERBB2 unamplified breast cancer. Lab Invest 2022, doi:10.1038/s41374-022-00804-9.

Reviewer 3 Report

The investigators evaluate associations of HER2 IHC status with RT-PCR-based (21-gene RS assay) HER2 expression levels in an ER+ breast cancer cohort, and clinically annotate these subsets. Some of the findings are interesting. For example, they find some differences in tumor characteristics according to HER2 IHC status (table 1-2), and they show differences in mean HER2 mRNA expression according to IHC strata (IHC 0, 1, 2, 3). They also attempt to use HER2 mRNA levels as means to predict IHC status (supplemental figures and tables), however the ROC analysis argues that they are poorly correlated. In the discussion section, they conclude that IHC 0+ and IHC 1+ tumor are similar, in comparison to the distinctive entity of IHC 2+.

However, putting it all together, I think most of the findings are modest and expected (i.e. the association of HER2 mRNA level with adverse prognostic features such as Ki67), whereas some of the findings are overstated. For example, in my opinion the data do not support the assertion that IHC 0 and IHC 1+ tumors are a distinct entity compared to IHC 2+. The differences in mRNA expression are subtle, and there seems to be an expected gradient in mRNA expression in IHC 0 v. 1 v. 2. The differences in clinical characteristics are subtle and the survival outcomes are similar. The differences in AUC for the ROC analysis for IHC 0 v. 1/2, versus IHC 0/1 v. 2, are nearly identical (AUC 0.6 v. 0.625). The difference in mRNA distribution across quantiles (table S2) is hard to discern by looking at the table, but differences seem modest.

Furthermore, I have difficulty understanding the clinical implications of these data. Antibody-drug conjugates presumptively work by binding to HER2 protein on the cell surface, so it does not make sense to propose using RNA expression as a biomarker for patient selection. Furthermore, in Destiny 04 we see excellent clinical activity of T-DxD in both IHC 1+ and IHC 2+ subsets, arguing against the conclusions of the authors that IHC 1+ is more similar to IHC 0+ versus IHC 2+.

 Before publication, the overall message and interpretation of the data needs to be reimagined.

Additional comments:

  1. Please clarify whether the RT-PCR was conducted as part of the OncotypeDx 21 gene recurrence score, or if it was done by another laboratory using a similar method
  2. Clarify how the HER2-positive comparator group in 2A was identified, and why they had the 21 gene RS score data, if genomic testing is not customarily obtained in this subgroup.
  3. Please clarify in manuscript why a low deltaCT score indicates a higher mRNA expression, it is not clear in the manuscript
  4. What is the definition of molecular subtype in table 2?
  5. Data in Table S2 would be better depicted graphically, as it is hard to quickly see the trend with the table format
  6. The AUC curves in figure s1 look very similar, with only very modest prediction for IHC status by mRNA level. I do not think this is compelling enough to argue that IHC 2+ is distinct from IHC 0/1+

Author Response

point-by-point response to the reviewer’s comments: 

Point 1:. I have difficulty understanding the clinical implications of these data. Antibody-drug conjugates presumptively work by binding to HER2 protein on the cell surface, so it does not make sense to propose using RNA expression as a biomarker for patient selection. Furthermore, in Destiny 04 we see excellent clinical activity of T-DxD in both IHC 1+ and IHC 2+ subsets, arguing against the conclusions of the authors that IHC 1+ is more similar to IHC 0+ versus IHC 2+.

Response 1: Thank you for your comment. The current diagnostic tests for HER2, ie. IHC and FISH, have been optimized for detecting HER2 positivity or negativity, while low concordance rates were found in the evaluation of HER2 IHC status in these patients with HER2 0 or HER2 1+ tumors[1,2]. Meanwhile, IHC is a semiquantitative assay and may be influenced by observer variability and other factors[3,4]. Moreover, previous studies reported that qRT-PCR as a quantitative assay had high concordance with IHC/FISH in assessing HER2 positive and negative status[3,5-7]. Thus, we attempted to look for whether qRT-PCR can examine HER2 level in this group of patients with HER2-negative tumors.

The mechanism of action of ADC requires certain expression of HER2 protein in tumor cells, while currently, the lower boundary of HER2-low expression was not clear[4]. As you have mentioned, T-Dxd showed promising anti-tumor activity in HER2 1+ and HER2 2+ subsets as reported by Destiny-breast 04. However, the DAISY (NCT04132960) trial reported that the antitumor activity of T-Dxd was significantly associated with HER2 expression level. The higher HER2 expression, the better response rate and PFS with T-Dxd. Meantime, they found that HER2 0 tumors also showed response to T-Dxd [8]. Therefore, the current definition of HER2-low expression with the lower boundary of HER2 IHC 1+ may be inaccurate. Thus, more precise assays to accurately detect HER2-low expression and define the lower boundary of HER2 low were essential, especially in the era of anti-HER2 ADC treatment.

Point 2:. Please clarify whether the RT-PCR was conducted as part of the OncotypeDx 21 gene recurrence score, or if it was done by another laboratory using a similar method.

Response 2: Thank you for your comment. The RT-PCR was conducted as part of a 21-gene recurrence assay using similar method to OncotypeDX. Our panel has been retrospectively validated in both node-negative and node-positive patients with two large cohorts of Chinese patients [9,10]. Our previous study indicated that poorly-differentiated, PR-negative, or high-proliferating tumors were significantly associated with a high RS score in early breast cancer patients, which was comparable to the findings from Oncotype Dx in the NSABP B-14 study population[11]. Meantime, the ER and PR expression level detected by IHC were highly linearly associated with their mRNA level by qRT-PCR testing. (We have clarified it in the manuscript as shown in the attached Word)

Point 3:. Clarify how the HER2-positive comparator group in 2A was identified, and why they had the 21 gene RS score data, if genomic testing is not customarily obtained in this subgroup.

Response 3: Thank you for your question. As you have mentioned, genomic testing with 21-gene RS is not a routine for HER2-positive patients. For these HER2-positive patients with 21-gene RS results, 21-gene RS testing was ordered under the circumstances as follows: HER2-positive with heterogeneity, or HER2 IHC 3+ with FISH-negative. In these circumstances, 21-gene RS testing would be recommended to the patients to test the HER2 mRNA level for reference, and to provide genetic data for adjuvant treatment decision. Therefore, in our Breast Cancer database, a very small proportion of HER2-positive breast tumor patients (n=48) had 21-gene RS score data and we include them in our study as positive control.

Point 4:. Please clarify in manuscript why a low delta CT score indicates a higher mRNA expression, it is not clear in the manuscript.

Response 4: Thank you for your comment. The CT (cycle threshold) valve is the number of amplification cycles, when the fluorescence signal of the amplification product reaches the set fluorescence threshold during the qRT-PCR amplification process. Thus, higher CT value means lower initial RNA levels and lower CT value means higher initial RNA levels. Every HER2 CT value was adjusted by five endogenous reference genes (β-actin, GAPDH, GUS, RPLPO, TFRC), and was recorded as ΔCT= CT HER2 – CT Reference[12]. Thus, a low delta CT value indicates a higher mRNA expression in our study.

Point 5:. What is the definition of molecular subtype in table 2?

Response 5: Thank you for your question. According to the 2013 St. Gallen International Expert Consensus, the definition of molecular subtype in table 2 were as follows:

Luminal A (ER+ / HER2 -, Ki-67<14%, and PR≥20%);

Luminal B- HER2 negative (ER+ / HER2 -, Ki-67≥14% or PR<20%).
(We have clarified it in the manuscript as shown in the attached Word)

Point 6:. Data in Table S2 would be better depicted graphically, as it is hard to quickly see the trend with the table format.

Response 6: Thank you for your suggestion. We have taken your advice and depicted the Table S2 into Figure S2. (as shown in the attached Word)

Point 7:. The AUC curves in figure s1 look very similar, with only very modest prediction for IHC status by mRNA level. I do not think this is compelling enough to argue that IHC 2+ is distinct from IHC 0/1+.

Response 7: Thank you for your comment. In our study, we firstly found that HER2 1+ cases were more similar to HER2 0 cases, rather than HER2 2+/FISH- cases in terms of clinicopathologic features (Table 1 and Table 2). Moreover, no statistically significant difference in HER2 mRNA expression was detected between HER2 0 and HER2 1+ cases, both lower than HER2 2+/FISH- cases (P<0.001, Figure 2). Thus, we assumed that HER2 0 and 1+ tumors may be distinct from HER2 2+/FISH- tumors.

We further attempted to explore whether HER2 mRNA level can be applied to distinguish HER2 status, so as to achieve a more precise HER2 classification than IHC. In our study, ROC curves were applied to determine optimal HER2 mRNA ΔCT cut-off. Then, we transform HER2 mRNA ΔCT from continuous variables to categorical variables by the optimal ΔCT cut-off value, and compared the concordance between mRNA level and IHC score. However, we found that categorized mRNA level and IHC score were discordant in HER2-negative tumors (Figure 3). We fully agree with you that there was only very modest prediction for IHC status by mRNA level. Meantime, we think that IHC was not a desirable testing to detect the lower boundary of HER-low expression. Also, the accuracy of HER2 mRNA level in determining HER2 expression status needed further researches.

In addition, 59.7% HER2 0/1+ tumors were mRNA high level (Figure 3B). Thus, we thought that it was controversial to consider HER2 1+ as the lower boundary of HER2-low expression. As a result, some of the HER2 0/1+ breast cancer patients might be assigned to wrong HER2 status, which may lead to some candidates not receiving appropriate treatment. Therefore, we thought that more accurate assays were necessary to evaluate HER2-low expression.

Round 2

Reviewer 1 Report

The authors gave detailed responses to the reviewer's comments and substantially revised the manuscript. I think that in its present form the article can be recommended for publication.

Author Response

Response:

Thank for all your comments and suggestions!

Reviewer 2 Report

Breast cancer is  widely accepted as a highly heterogeneous disease.  In the clinic, the Her2-low or negative tumors  are also subjected to analysis of ER and PR expression. As a result of such analysis, the Her2-negative tumors are  readily divided into  ER+ and triple-negative subtypes., which are also still under intensive mechanistic  studies and testing of targeted therapies. Along this line, more study on the trivial differences in Her2 expression may have limited contribution to our pathological understanding  and treatment of these grossly regarded her2-negative breast tumors.  

Author Response

Response to the comment:

Thank you for your comment.

We agree with you that breast cancer is a highly heterogeneous disease. In our point of view, it is important to look into HER2 expression, and to tell HER2 low tumors from HER2 truly negative tumors. Our reasons are as follows:

First, HER2-low tumor is a new entity defined as HER2 1+ or 2+ with FISH negative, which has been reported distinct from HER2 0 tumors in terms of clinicopathologic characteristics, molecular features, and survival [1-3].

Next, current HER2 testing algorithm is initially designed to distinguish “HER2 positive” from “HER2 negative” [4-6]. Thus, the upper boundary of HER2-low is widely-accepted, which is IHC 2+ FISH-negative, but the lower boundary of HER2-low expression remains undetermined. Fernandez et al. have reported a low concordance rate of 26% between HER2 0 and 1+ tumors when these cases were evaluated by different pathologists [4].

Moreover, recent studies have reported that HER2-low breast cancer can benefit from treatment with HER2-targeted antibody-drug conjugates (ADCs) [7,8] , such as trastuzumab deruxtecan (T-Dxd) [9] and trastuzumab duocarmazine (SYD985) [10]. FDA has recently approved T-Dxd for the treatment in patients with unresectable or metastatic HER2-low breast cancer on August 5, 2022 [11]. As a result, to distinguish HER2-low tumor from HER2 truly negative tumor, has therapeutic significance.

In a word, it is of clinical implication to explore the precise definition for HER2-low expression, which may help precisely select candidates for anti-HER2 ADCs. This is the main aim of our study.

In our study, we found that HER2 0 and HER2 1+ cases shared great similarities in terms of clinicopathological features, which were differently distributed compared with HER2 2+/FISH-negative cases. More importantly, our study demonstrated that the HER2 mRNA expression levels were similar between HER2 0 and HER2 1+ cases, both lower than HER2 2+/FISH-negative cases, indicating current lower boundary of HER2-low expression evaluated by IHC/FISH may be inappropriate. More studies are necessary to clarify the definition of HER2-low expression.

Reference

  1. Denkert, C.; Seither, F.; Schneeweiss, A.; Link, T.; Blohmer, J.U.; Just, M.; Wimberger, P.; Forberger, A.; Tesch, H.; Jackisch, C.; et al. Clinical and molecular characteristics of HER2-low-positive breast cancer: pooled analysis of individual patient data from four prospective, neoadjuvant clinical trials. Lancet Oncol 2021, 22, 1151-1161, doi:10.1016/S1470-2045(21)00301-6.
  2. Zhang, G.; Ren, C.; Li, C.; Wang, Y.; Chen, B.; Wen, L.; Jia, M.; Li, K.; Mok, H.; Cao, L.; et al. Distinct clinical and somatic mutational features of breast tumors with high-, low-, or non-expressing human epidermal growth factor receptor 2 status. BMC Med 2022, 20, 142, doi:10.1186/s12916-022-02346-9.
  3. Schettini, F.; Chic, N.; Braso-Maristany, F.; Pare, L.; Pascual, T.; Conte, B.; Martinez-Saez, O.; Adamo, B.; Vidal, M.; Barnadas, E.; et al. Clinical, pathological, and PAM50 gene expression features of HER2-low breast cancer. NPJ Breast Cancer 2021, 7, 1, doi:10.1038/s41523-020-00208-2.
  4. Fernandez, A.I.; Liu, M.; Bellizzi, A.; Brock, J.; Fadare, O.; Hanley, K.; Harigopal, M.; Jorns, J.M.; Kuba, M.G.; Ly, A.; et al. Examination of Low ERBB2 Protein Expression in Breast Cancer Tissue. JAMA Oncol 2022, 8, 1-4, doi:10.1001/jamaoncol.2021.7239.
  5. Ruschoff, J.; Lebeau, A.; Kreipe, H.; Sinn, P.; Gerharz, C.D.; Koch, W.; Morris, S.; Ammann, J.; Untch, M.; Nicht-interventionelle Untersuchung, H.E.R.S.G. Assessing HER2 testing quality in breast cancer: variables that influence HER2 positivity rate from a large, multicenter, observational study in Germany. Mod Pathol 2017, 30, 217-226, doi:10.1038/modpathol.2016.164.
  6. Pfitzner, B.M.; Lederer, B.; Lindner, J.; Solbach, C.; Engels, K.; Rezai, M.; Dohnal, K.; Tesch, H.; Hansmann, M.L.; Salat, C.; et al. Clinical relevance and concordance of HER2 status in local and central testing-an analysis of 1581 HER2-positive breast carcinomas over 12 years. Mod Pathol 2018, 31, 607-615, doi:10.1038/modpathol.2017.171.
  7. Tarantino, P.; Hamilton, E.; Tolaney, S.M.; Cortes, J.; Morganti, S.; Ferraro, E.; Marra, A.; Viale, G.; Trapani, D.; Cardoso, F.; et al. HER2-Low Breast Cancer: Pathological and Clinical Landscape. J Clin Oncol 2020, 38, 1951-1962, doi:10.1200/JCO.19.02488.
  8. Eiger, D.; Agostinetto, E.; Saude-Conde, R.; de Azambuja, E. The Exciting New Field of HER2-Low Breast Cancer Treatment. Cancers (Basel) 2021, 13, doi:10.3390/cancers13051015.
  9. Modi, S.; Park, H.; Murthy, R.K.; Iwata, H.; Tamura, K.; Tsurutani, J.; Moreno-Aspitia, A.; Doi, T.; Sagara, Y.; Redfern, C.; et al. Antitumor Activity and Safety of Trastuzumab Deruxtecan in Patients With HER2-Low-Expressing Advanced Breast Cancer: Results From a Phase Ib Study. J Clin Oncol 2020, 38, 1887-1896, doi:10.1200/JCO.19.02318.
  10. Banerji, U.; van Herpen, C.M.L.; Saura, C.; Thistlethwaite, F.; Lord, S.; Moreno, V.; Macpherson, I.R.; Boni, V.; Rolfo, C.; de Vries, E.G.E.; et al. Trastuzumab duocarmazine in locally advanced and metastatic solid tumours and HER2-expressing breast cancer: a phase 1 dose-escalation and dose-expansion study. Lancet Oncol 2019, 20, 1124-1135, doi:10.1016/S1470-2045(19)30328-6.
  11. FDA Approves First Targeted Therapy for HER2-Low Breast Cancer. Available online: https://www.fda.gov/news-events/press-announcements/fda-approves-first-targeted-therapy-her2-low-breast-cancer (accessed on August 08, 2022)

Reviewer 3 Report

Point 1:

- i think more work is needed to develop a stronger case for how the data are clinically relevant, how they can be reconciled with data from destiny and daisy. the conclusions and interpretation of data need to be rewritten cautiously. For example, i respectfully disagree with the conclusion in the simple summary, that that HER2 1+ disease should not be considered HER2-low.  we know from the Destiny 04 that patients with IHC 1+ benefited from TDxD. 

Point 2:

- thank you. is it the same formula and genes as oncotype? what does "similar" mean

Point 3: 

- selection of HER2-positive patients seems to be biased, as the authors point out in comments. this must be explained in manuscript and described as a caveat

Point 4: 

-thank you, can you please explain this in manuscript, to improve readability?

Point 5: 

-thank you!

Point 6:

-thank you! it looks pretty but i think would be easier to see differences across the subgroups if the distributions were overlaid, or stacked vertically, rather than horizontally. they all look the same in this figure. but yes this is nicer than table thank you. The differences look very subtle so you need to the best you can to highlight the differences. may i suggest converting  the histogram into a smoothed density plot/curve, and overlay the density plots for each of the subgroups, so you can see where the differences are in the density distributions?

Point 7

- I am very confused as to why you would be trying to predict IHC results using mRNA. isn't this at odds with your argument that IHC is not a good biomarker? i think a more relevant experiment would be to see whether mRNA predicts clinical outcome following treatment with TDxD.  

- i think the point you are trying to make with the ROC analysis is that mRNA does not correlate well with IHC for IHC 0-2+ cases? 

Additional point:

- table 2:  you are arguing that HER2 0+ are 1+ are similar, whereas HER2 2+ different. therefore, for table 2 could you also repeat the analysis looking at HER2 1+ versus HER2 2+ (i.e. to add on page 5)? would be nice to include this in table 2 as well, do all pair-wise comparisons. 

Author Response

Response to comments:

Point 1:

- i think more work is needed to develop a stronger case for how the data are clinically relevant, how they can be reconciled with data from destiny and daisy. the conclusions and interpretation of data need to be rewritten cautiously. For example, i respectfully disagree with the conclusion in the simple summary, that that HER2 1+ disease should not be considered HER2-low.  we know from the Destiny 04 that patients with IHC 1+ benefited from TDxD. 

Response 1:

Thank you for your comment. We totally agreed with you that it was inappropriate to simply think HER2 1+ tumors should not be considered as HER2-low tumors. We have made revisions in the last paragraph “5. Conclusions”.

As you have mentioned, T-Dxd showed promising anti-tumor activity in HER2 1+ and HER2 2+ subsets as reported by Destiny-breast 04. However, the DAISY (NCT04132960) trial reported that the antitumor activity of T-Dxd was significantly associated with HER2 expression level. The higher HER2 expression, the better response rate and PFS with T-Dxd. Meantime, they found that HER2 0 tumors also showed response to T-Dxd [1].

Combined with our conclusions, HER2 1+ tumors were more similar to HER2 0 tumors, which have diverse clinicopathologic factors and HER2 mRNA levels compared with HER2 2+/FISH- cases. Therefore, part of patients with HER2 0 or HER2 1+ tumors might be assigned the wrong HER2 status. The current definition of HER2-low expression with the lower boundary of HER2 IHC 1+ may be inaccurate. The true HER2 status in HER2 1+ and HER2 0 tumors may need further evaluation, which may help to better predict prognosis and response to T-Dxd.

We have made revisions in the manuscript accordingly, as follows:

  1. Conclusions

Our study found that HER2-low patients with HER2 1+ tumors were more similar to HER2 0 tumors, which have diverse clinicopathologic factors and HER2 mRNA levels compared with HER2 2+/FISH- cases, indicating the current definition of HER2-low expression with the lower boundary of HER2 IHC 1+ may be inaccurate. Either IHC or qRT-PCR may not be the optimal assay to quantify HER2-low population. Trials integrating quantitative qRT-PCR testing of HER2 mRNA levels and response to anti-HER2 ADCs treatment are warranting further evaluation.

Point 2:

- thank you. is it the same formula and genes as oncotype? what does "similar" mean.

Response 2:

Thank you for your comment. The RT-PCR was conducted as part of a 21-gene recurrence assay using same formula and genes to OncotypeDX.

We have made revisions in the manuscript accordingly, as follows:

2.3. HER2 mRNA expression by qRT-PCR

The expression of HER2 mRNA was measured by qRT-PCR as part of a center-specific 21-gene recurrence (RS) score testing assay using the same formula and genes to OncotypeDX. This RT-PCR-based panel and has been validated in large cohorts of Chinese patients and has been routinely applied to guide adjuvant treatment and predict disease outcomes since 2015 [2,3]. In brief, three 10-μm unstained slides of formalin-fixed, paraffin-embedded (FFPE) breast tumor specimens were prepared and reviewed by the pathologists to ensure that invasive tumor components accounted for at least 50% of the section. RNA was extracted using an RNeasy FFPE kit (Qiagen, 73504). Subsequently, reverse transcription was performed using an OmniScript RT kit (Qiagen 205111). PCR was accomplished in the Applied Biosystems 7500 Real-Time PCR System (Foster City, CA) with Premix Ex TaqTM (TaKaRa Bio, RR390A). The expression of HER2 mRNA was measured by cycle threshold (CT), adjusted by five endogenous reference genes β-actin, GAPDH, GUS, RPLPO, TFRC, and was recorded as ΔCT= CT HER2 – CT Reference.

Point 3: 

- selection of HER2-positive patients seems to be biased, as the authors point out in comments. this must be explained in manuscript and described as a caveat.

Response 3:

Thank you for your advice.

We have clarified it in the manuscript as follows:

2.3. HER2 mRNA expression by qRT-PCR

The expression of HER2 mRNA was measured by qRT-PCR as part of a center-specific 21-gene recurrence score (RS) testing assay using the same formula and genes to OncotypeDX. This RT-PCR-based panel and has been validated in large cohorts of Chinese patients and has been routinely applied to guide adjuvant treatment and predict disease outcomes since 2015 [2,3]. In brief, three 10-μm unstained slides of formalin-fixed, paraffin-embedded (FFPE) breast tumor specimens were prepared and reviewed by the pathologists to ensure that invasive tumor components accounted for at least 50% of the section. RNA was extracted using an RNeasy FFPE kit (Qiagen, 73504). Subsequently, reverse transcription was performed using an OmniScript RT kit (Qiagen 205111). PCR was accomplished in the Applied Biosystems 7500 Real-Time PCR System (Foster City, CA) with Premix Ex TaqTM (TaKaRa Bio, RR390A). The expression of HER2 mRNA was measured by cycle threshold (CT), adjusted by five endogenous reference genes β-actin, GAPDH, GUS, RPLPO, TFRC, and was recorded as ΔCT= CT HER2 – CT Reference. Forty-eight patients with heterogeneous HER2-positive disease, or HER2 IHC 3+ with FISH-negative were considered HER2-positive, and included as positive control. In these circumstances, 21-gene RS testing would be recommended to test the HER2 mRNA level for reference, and to provide genetic data to guide adjuvant treatment decision.

  1. Discussion

Several limitations of our study require attention. Firstly, the study was retrospectively designed and carried out in a single-center, which could cause certain selection bias. Additionally, forty-eight patients with heterogeneous HER2-positive disease, or HER2 IHC 3+ with FISH-negative were considered HER2-positive, and included in our study as positive control. These HER2-positive patients were not randomly selected, which might cause selection bias. And then, only HR-positive HER2-negative breast cancer patients were included for HER2-low population analysis in this cohort. However, in the HER2-low breast tumors, it has been reported that HR-negative cases have lower HER2 mRNA levels than HR-positive cases[4]. Moreover, a phase Ib clinical trial of T-Dxd reported that the ORR seemed to be different according to HR status (HR-positive 40.4% vs. HR-negative 14.3%)[5]. Therefore, in HR-negative cases, further researches are deserved to explore the characteristics of HER2-low tumors and to find out whether HER2 1+ and HER2 2+/FISH- tumors can be considered an entity.

Point 4: 

-thank you, can you please explain this in manuscript, to improve readability?

Response 4:

Thank you for your suggestion.

We have clarified the definition of CT in the manuscript as follows:

2.3. HER2 mRNA expression by qRT-PCR

The expression of HER2 mRNA was measured by qRT-PCR as part of a center-specific 21-gene recurrence score (RS) testing assay using the same formula and genes to OncotypeDX. This RT-PCR-based panel and has been validated in large cohorts of Chinese patients and has been routinely applied to guide adjuvant treatment and predict disease outcomes since 2015 [2,3]. In brief, three 10-μm unstained slides of formalin-fixed, paraffin-embedded (FFPE) breast tumor specimens were prepared and reviewed by the pathologists to ensure that invasive tumor components accounted for at least 50% of the section. RNA was extracted using an RNeasy FFPE kit (Qiagen, 73504). Subsequently, reverse transcription was performed using an OmniScript RT kit (Qiagen 205111). PCR was accomplished in the Applied Biosystems 7500 Real-Time PCR System (Foster City, CA) with Premix Ex TaqTM (TaKaRa Bio, RR390A). The expression of HER2 mRNA was measured by cycle threshold (CT). The CT valve was the number of amplification cycles during the qRT-PCR amplification process, when the fluorescence signal of the amplification product reached the set fluorescence threshold. Thus, higher CT value means lower initial RNA levels and lower CT value means higher initial RNA levels. HER2 CT value was adjusted by five endogenous reference genes (β-actin, GAPDH, GUS, RPLPO, TFRC), and was recorded as ΔCT= CT HER2 – CT Reference[6]. Forty-eight patients with heterogeneous HER2-positive disease, or HER2 IHC 3+ with FISH-negative were considered HER2-positive, and included as positive control. In these circumstances, 21-gene RS testing would be recommended to test the HER2 mRNA level for reference, and to provide genetic data to guide adjuvant treatment decision.

Point 5:

-thank you! it looks pretty but i think would be easier to see differences across the subgroups if the distributions were overlaid, or stacked vertically, rather than horizontally. they all look the same in this figure. but yes this is nicer than table thank you. The differences look very subtle so you need to the best you can to highlight the differences. may i suggest converting the histogram into a smoothed density plot/curve, and overlay the density plots for each of the subgroups, so you can see where the differences are in the density distributions?

Response 5:

Thank you for your suggestion. We have taken your advice and converted the histogram into a smoothed density plot. As Figure S2 showed, significant overlaps were observed between HER2 0 and HER2 1+ groups, relatively distinct from HER2 2+/FISH- cases.

We have made revisions in the manuscript accordingly, as follows (Figure S2 as shown in the attached Word):

2.5. Statistical analysis

Categorical data were analyzed using chi-squared test (Fisher’s exact test when necessary) and multinomial logistic regression analysis. The Kruskal-Wallis test was applied to compare HER2 mRNA expression levels by different HER2 IHC/FISH statuses. Receiver operator characteristic (ROC) curves were used for obtaining the optimal ΔCT cut-off value at the most extensive Youden index. Cohen’s kappa statistics were used for evaluating concordance between IHC/FISH and mRNA level by qRT-PCR in determining HER2 status. The comparison of continuous HER2 mRNA level among different HER2 IHC/FISH statuses were shown in smoothed density plot. Survival curves were performed by Kaplan-Meier analysis and log-rank test. All statistical analyses were carried out by SPSS version 24.0 (SPSS, Inc, Chicago, IL), GraphPad Prism9.0 (GraphPad Software, San Diego) and R software (version 4.0.5). All statistical tests were two-tailed and were considered statistically significant at P values < 0.05.

3.2. HER2 mRNA expression comparison by HER2 IHC/ FISH status

HER2 mRNA expression was determined by qRT-PCR and recorded as ΔCT (CTHER2- CT reference). The medians of ΔCT were 2.94 (N=368; Q1-Q3: 2.25-3.57), 2.61 (N=1928; Q1-Q3: 1.90-3.23) and 1.18 (N=48; Q1-Q3: 0.24-1.90) for HER2 0, HER2-low, and HER2-positive cases (Figure 2A). HER2 positive cases showed the highest mRNA expression level while HER2 0 showed the lowest mRNA expression level, and all pairwise comparisons were significantly different (P<0.001; Figure 2A). When further stratified by IHC score in HER2-low tumors, the medians of ΔCT were 2.80 (N=911; Q1-Q3: 2.16-3.46) and 2.47 (N=1017; Q1-Q3: 1.74-3.03) for HER2 1+ and HER2 2+/FISH- cases (Figure 2B), respectively. We found that HER2 mRNA expression of HER2 1+ cases was similar to that of HER2 0 cases (P=0.063), both much less than HER2 2+/FISH- cases (P<0.001; Figure 2B). Significant overlaps were shown between HER2 0 and HER2 1+ groups, relatively distinct from HER2 2+/FISH- group (Figure S2). 

Point 6

- I am very confused as to why you would be trying to predict IHC results using mRNA. isn't this at odds with your argument that IHC is not a good biomarker? i think a more relevant experiment would be to see whether mRNA predicts clinical outcome following treatment with TDxD.  

- i think the point you are trying to make with the ROC analysis is that mRNA does not correlate well with IHC for IHC 0-2+ cases? 

Response 6:

Thank you for your comment. Yes, what we tried to make with ROC analysis was to explore whether mRNA correlated well with IHC for IHC 0-2+ cases. Previous studies reported that qRT-PCR as a quantitative assay had high concordance with IHC/FISH in assessing HER2 positive and negative status[7-10]. However, we found that categorized mRNA level and IHC status were discordant in HER2-negative tumors (Figure 3), indicating that the mRNA level might not be in linear relation to HER2 IHC status in HER2-negative tumors.

We totally agree with you that a more relevant experiment should be to see whether mRNA could predict clinical outcome following treatment with T-Dxd. We hope that future clinical trials will look into whether mRNA level could be a better marker of efficacy to T-Dxd.

Additional point:

- table 2:  you are arguing that HER2 0+ are 1+ are similar, whereas HER2 2+ different. therefore, for table 2 could you also repeat the analysis looking at HER2 1+ versus HER2 2+ (i.e. to add on page 5)? would be nice to include this in table 2 as well, do all pair-wise comparisons. 

Response 7:

Thank you for your advice, we fully agree with your opinion that analysis should be repeated for HER2 1+ versus HER2 2+. Thus, we have taken your advice and made revisions in Table 2. We have changed the reference from HER2 0 category to HER2 2+/FISH- category in table 2, and done all pair-wise comparisons among HER2 0, HER2 1+ and HER2 2+/FISH- cases (Table 2).

We have made revisions in the manuscript accordingly, as follows:

Histology, TNM stage, tumor size, node status, PR, Ki-67, and molecular subtype were differently distributed among HER2 0, HER2 1+, and HER2 2+/FISH- in the univariate model (All P<0.05; Table 1). Multinomial logistic regression showed a significantly distinguishable distribution of histology (P=0.015), node status (P=0.016), and Ki-67 (P=0.025) among HER2 0, HER2 1+, and HER2 2+/FISH- cases (Table 2). Compared with patients in the HER2 2+/FISH- group, HER2 0 patients were more often non-IDC (odds ratio [OR] 1.57, 95% confidence interval [CI] 1.14-2.15, P=0.006), node-negative (OR 1.58, 95%CI 1.11-2.25, P=0.012), PR-negative (OR 1.54, 95%CI 1.08-2.20, P=0.017), when HER2 1+ patients were more often non-IDC ( OR 1.30, 95% CI 1.01-1.67, P=0.045), node-negative (OR 1.29, 95%CI 1.01-1.65, P=0.044), and more likely to have Ki-67<14% (OR 1.42, 95%CI 1.18-1.70, P<0.001). However, there was no statistically significant difference between HER2 0 and HER2 1+ group in terms of the above clinicopathologic features (All P>0.05).

Reference

  1. Dieras, V.; Deluche, E.; Lusque, A.; Pistilli, B.; Bachelot, T.; Pierga, J.Y.; Viret, F.; Levy, C.; Salabert, L.; Le Du, F.; et al. Trastuzumab deruxtecan (T-DXd) for advanced breast cancer patients (ABC), regardless HER2 status: A phase II study with biomarkers analysis (DAISY). Cancer Res 2022, 82, doi:10.1158/1538-7445.Sabcs21-Pd8-02.
  2. Wu, J.; Fang, Y.; Lin, L.; Fei, X.; Gao, W.; Zhu, S.; Zong, Y.; Chen, X.; Huang, O.; He, J.; et al. Distribution patterns of 21-gene recurrence score in 980 Chinese estrogen receptor-positive, HER2-negative early breast cancer patients. Oncotarget 2017, 8, 38706-38716, doi:10.18632/oncotarget.16313.
  3. Wu, J.; Gao, W.; Chen, X.; Fei, C.; Lin, L.; Chen, W.; Huang, O.; Zhu, S.; He, J.; Li, Y.; et al. Prognostic value of the 21-gene recurrence score in ER-positive, HER2-negative, node-positive breast cancer was similar in node-negative diseases: a single-center study of 800 patients. Front Med 2021, 15, 621-628, doi:10.1007/s11684-020-0738-0.
  4. Schettini, F.; Chic, N.; Braso-Maristany, F.; Pare, L.; Pascual, T.; Conte, B.; Martinez-Saez, O.; Adamo, B.; Vidal, M.; Barnadas, E.; et al. Clinical, pathological, and PAM50 gene expression features of HER2-low breast cancer. NPJ Breast Cancer 2021, 7, 1, doi:10.1038/s41523-020-00208-2.
  5. Modi, S.; Park, H.; Murthy, R.K.; Iwata, H.; Tamura, K.; Tsurutani, J.; Moreno-Aspitia, A.; Doi, T.; Sagara, Y.; Redfern, C.; et al. Antitumor Activity and Safety of Trastuzumab Deruxtecan in Patients With HER2-Low-Expressing Advanced Breast Cancer: Results From a Phase Ib Study. J Clin Oncol 2020, 38, 1887-1896, doi:10.1200/JCO.19.02318.
  6. Tong, Y.; Chen, X.; Fei, X.; Lin, L.; Wu, J.; Huang, O.; He, J.; Zhu, L.; Chen, W.; Li, Y.; et al. Can breast cancer patients with HER2 dual-equivocal tumours be managed as HER2-negative disease? Eur J Cancer 2018, 89, 9-18, doi:10.1016/j.ejca.2017.10.033.
  7. Gong, Y.; Yan, K.; Lin, F.; Anderson, K.; Sotiriou, C.; Andre, F.; Holmes, F.A.; Valero, V.; Booser, D.; Pippen, J.E., Jr.; et al. Determination of oestrogen-receptor status and ERBB2 status of breast carcinoma: a gene-expression profiling study. Lancet Oncol 2007, 8, 203-211, doi:10.1016/S1470-2045(07)70042-6.
  8. Wasserman, B.E.; Carvajal-Hausdorf, D.E.; Ho, K.; Wong, W.; Wu, N.; Chu, V.C.; Lai, E.W.; Weidler, J.M.; Bates, M.; Neumeister, V.; et al. High concordance of a closed-system, RT-qPCR breast cancer assay for HER2 mRNA, compared to clinically determined immunohistochemistry, fluorescence in situ hybridization, and quantitative immunofluorescence. Lab Invest 2017, 97, 1521-1526, doi:10.1038/labinvest.2017.93.
  9. Baehner, F.L.; Achacoso, N.; Maddala, T.; Shak, S.; Quesenberry, C.P., Jr.; Goldstein, L.C.; Gown, A.M.; Habel, L.A. Human epidermal growth factor receptor 2 assessment in a case-control study: comparison of fluorescence in situ hybridization and quantitative reverse transcription polymerase chain reaction performed by central laboratories. J Clin Oncol 2010, 28, 4300-4306, doi:10.1200/JCO.2009.24.8211.
  10. Noske, A.; Loibl, S.; Darb-Esfahani, S.; Roller, M.; Kronenwett, R.; Muller, B.M.; Steffen, J.; von Toerne, C.; Wirtz, R.; Baumann, I.; et al. Comparison of different approaches for assessment of HER2 expression on protein and mRNA level: prediction of chemotherapy response in the neoadjuvant GeparTrio trial (NCT00544765). Breast Cancer Res Treat 2011, 126, 109-117, doi:10.1007/s10549-010-1316-y.
